# African Leafy Vegetables for Improved Human Nutrition and Food System Resilience in Southern Africa: A Scoping Review

**Admire Isaac Tichafa Shayanowako** [1,2,*], **Oliver Morrissey** [3], **Alberto Tanzi** [4], **Maud Muchuweti** [5], **Guillermina M. Mendiondo** [4], **Sean Mayes** [4], **Albert T. Modi** [1] and **Tafadzwanashe Mabhaudhi** [1,*]

1   Centre for Transformative Agricultural and Food Systems, School of Agricultural, Earth and Environmental Sciences, University of KwaZulu-Natal, Pietermaritzburg 3209, South Africa; modiat@ukzn.ac.za
2   African Centre for Crop Improvement, School of Agricultural, Earth and Environmental Sciences, University of KwaZulu-Natal, Pietermaritzburg 3209, South Africa
3   School of Economics, University of Nottingham, Nottingham NG7 2RD, UK; Oliver.Morrissey@nottingham.ac.uk
4   Plant and Crop Sciences, School of Biosciences, University of Nottingham, Nottingham NG7 2RD, UK; Alberto.Tanzi1@nottingham.ac.uk (A.T.); Guillermina.Mendiondo@nottingham.ac.uk (G.M.M.); sean.mayes@nottingham.ac.uk (S.M.)
5   Department of Nutrition Dietetics and Food Science, Faculty of Science, University of Zimbabwe, 630 Churchill Avenue, Harare 0000, Zimbabwe; muchuweti@medic.uz.ac.zw
*   Correspondence: shayanowako@gmail.com (A.I.T.S.); mabhaudhi@ukzn.ac.za (T.M.)

**Abstract:** The economic potential of African leafy vegetables (ALVs) remains obscured by a poorly developed value chain. This scoping review assembled and examined scattered knowledge generated on ALVs across southern Africa, focusing on production, processing, marketing, and consumption. Two electronic databases (Scopus and Web of Science) were screened, and a total of 71 relevant studies were included and evaluated. The review provides a state of the art on knowledge related to utilisation of ALVs across the entire value chain. The findings show that functional properties are of prime importance in the production and consumption of ALVs. However, the lack of improved germplasm and a non-existent seed supply system are significant production bottlenecks. Pests and diseases affecting the productivity of ALVs remain mostly unexplored. Sun-drying and boiling were the most reported post-harvest processing methods, suggesting that traditional processing methods are still prominent. Many studies also confirmed the predominance of informal markets in the trading of ALVs as they fail to penetrate formal markets because of poor product positioning and exclusion from produce demand and supply forecasts. The inception of cultivar development, mechanised processing methods, and market linkages will enhance the profitability of ALVs in the region. This review enhances the gaining of insight into the state of different value chain components will assist in upscaling production, value addition of products, and enhance marketing efficiency. There is a great opportunity for basic and applied research into ALVs.

**Keywords:** food and nutrition security; indigenous; sustainability; underutilised; value chain

## 1. Introduction

Indigenous African leafy vegetables (ALVs) are an integral constituent of the diets of many southern Africans. The region is endowed with a vast array of (ALVs) with remarkably high concentrations of essential vitamins and minerals [1]. African leafy vegetables consist of all categories of plants whose leaves are acceptable and used as vegetables by rural and urban communities through tradition. Over 100 ALVs species have been reported across southern Africa [2]. *Amaranthus* spp., *Solanum* spp., *Cleome gynandra*, *Vigna unguiculata*, *Bidens pilosa*, and *Corchorus olitorius* are among the cultivated ALVs [3]. African leafy vegetables are rich sources of essential micronutrients such as vitamins A, B, and C and minerals like calcium, iron, zinc, selenium, copper, and potassium. A 100 g servings of

cleome, *Solanum nigram*, and *Amaranthus* contain more than double the concentration of amino acids, beta-carotene, and iron than *Brassica oleracea* var. *capitate* [4], which are widely consumed by urban populations. Moreover, ALVs are ideal for subsistence because they can be harvested frequently, in small quantities, and without the need for storage [5]. They are also closely tied to cultural traditions, and therefore have an essential role in supporting social diversity. The use of these plants will also contribute to the sustainable management of fragile ecosystems.

ALVs such as the *Biden pilosa*, *Amaranthus*, *Cleome gynandra*, *Cucurbitaceae* spp. (leaf), and *Ipomoea batatas* (leaf) have ratoon ability coupled with the ability to grow all year round, especially in warm subtropical environments. Some ALVs such as the *cucurbits* have rapid canopy expansion rates, which allow them to accumulate vegetative biomass within a short period, even before the harvest of the earliest maturing legumes. This guarantees food and nutrition security before the harvest of cereals and legume crops. Since ALVs are indigenous to the African continent, they have undergone many generations of selection against multiple stresses associated with tropical environments, particularly drought. Hence, they have co-evolved adaptive mechanisms to ensure broad adaptation. It is noteworthy that each ALV has adaptive features that promote growth under marginal conditions where most exotic vegetables fail. For instance, *Amaranthus* spp. has leaves with a waxy cuticle, which protects against rapid moisture loss [6]. African leafy vegetables such as *Amaranthus*, *Brassica nigra*, and *Cleome gynandra* are also drought hardy because of their excellent stomatal conductance. Some ALVs species such as *Bidens pilosa* have a remarkable recovery rate after exposure to a prolonged drought. Most ALVs come into production within a very short time after the first rains, and harvesting commences three to four weeks after emergence, while few ALVs require high amounts of fertiliser and pesticides, unlike most exotic vegetables such as *Brassica* spp. that are renowned for being heavy feeders of nitrogen. Hence, promoting their cultivation aligns with sustainable agricultural practices

Despite the potential of ALVs to contribute to food and nutrition security, the ALVs remain classified as underutilised crop species [6,7]. Hence ALVs lack a defined product value chain in the entire southern African region [8]. Production of ALVs is still concentrated on marginal lands as the best resources are dedicated to producing the main cereals, legumes, and a few exotic vegetables. Poor seed quality remains a major constraint limiting the productivity of ALVs as farmers still rely on retained seeds of landraces because the formal seed retailers have been slow to enter this market gap [9]. The lack of improved cultivars and continued use of landraces means that productivity remains low (yield per unit land) [1]. Hence, farmers growing ALVs devote more effort to other higher-yield popular crops, including exotic vegetables. Supply chains of ALVs are also yet to be defined to meet the growing demand for ALVs in urban communities. Currently, informal urban markets are the major suppliers of ALVs, and women take the lead in both production and marketing of indigenous vegetables [1]. Where ALVs have been commercialised, middle salespersons purchase the vegetables from rural farmers for resale in urban markets at retail prices [10]. Hence, policymakers need to develop a product value chain for the ALVs as the bulk of the trade is restricted to informal markets. The development of demand-oriented value chains is driven by empirical studies that reveal current knowledge, opportunities, and constraints affecting the successful transformation of ALV businesses in Africa.

The past and present research has focused on the value chain's start-up point with limited focus on the value chain's supporting activities. It suffices to say that sufficient awareness has been raised on the potential functional benefits of ALVs [1,10–14]. Further efforts have also been directed at the phenotypic characterisation of ALVs species [15,16]. While these efforts have led to a better comprehension of the impact of ALVs in the mitigation of hunger and malnutrition, less information is available on how to transform them into commercial crops [17]. According to [18], ALVs remain underutilised due to constraints such as perception, processing, distribution, marketing, and lack of information on their nutritional value bioavailability. There is a need to devise a strategy for upscaling ALVs while focusing on positioning them towards building resilience against climate

change-induced droughts on food and nutrition security for rural and urban communities. The foremost step involves a systematic diagnosis of the challenges, obstacles, and opportunities and developing recommendations for dealing with the challenges.

Researchers have used scoping review methodologies to systematically map the literature on a topic to identify the key concepts, existing evidence, and gaps in an existing research theme [19]. The scoping approach is designed to serve in emerging fields such as those involving underutilised crop species with a paucity of information from well-replicated trials from which conclusive findings may be used to guide decision making. A lot is known about the rich nutritional value of various ALVs, but less is known about their economic value. Hence this study will be the first attempt to employ the scoping review method to identify the existing knowledge, gaps, and evidence useful for priority setting in the upscaling of AVLs in southern Africa. To the best of our knowledge, this is the first state-of-the-art review and knowledge synthesis on the status of utilisation for ALVs from a value chain perspective.

## 2. Materials and Methods

The scoping review methodology proposed by [19] offers a rigorous and comprehensive process for synthesising research evidence. This study follows the five stages, which include: (1) Identifying the research question, (2) identifying relevant studies, (3) study selection (4) charting data, and (5) collating, summarising, and reporting the results [19]. In the current study, the aim was to reveal the benefits and constraints affecting the ALV value chain's upscaling in southern Africa. The study synthesised research evidence relating to the current knowledge on production risks and willingness to adopt new ALV varieties, the agronomic performance of various ALVs; market demand, consumption, and nutrition. Hence, a comprehensive synthesis of peer-reviewed articles and grey literature was done.

### 2.1. Research Question

Scoping study research questions are broad as the focus is on summarising the breadth of evidence. According to [19], there is a need to formulate a research question that has a broad scope, enough to conform to all the subsequent stages of the research process, such as identifying studies and making decisions about study inclusion. Here, the following research question was key 'What components of the ALVs value-chain exist in southern Africa?'. The current study used the target population, interventions, and outcomes of interest to the scoping study to establish an adequate study design. The population was specified as breeders, farmers, traders, and consumers of ALVs. The intervention involved the production, processing, marketing, and consumption of ALVs. The outcome was defined as the establishment of a viable product value chain for ALVs. Study designs were categorised into quantitative, qualitative, and mixed methods.

### 2.2. Identification of Relevant Studies, Data Sources, and Search Strategy

Scopus and Web of Science (WoS) were used to assemble publications on ALVs in southern Africa. In this particular study, southern Africa was defined based on the main-land Southern Africa Development Committee (SADC) states comprising of Angola, Botswana, the Democratic Republic of Congo, Eswatini, Lesotho, Malawi, Mozambique, Namibia, South Africa, Tanzania, Zambia, and Zimbabwe. The island states (Comoros, Madagascar, Mauritius, and Seychelles) were excluded from the study because they present a different context regarding agrobiodiversity and climate. These two search engines were chosen because of their ability to identify relevant publications based on complex search terms. The searches were limited to the English language in the publication years 1994 to 2020 (the search was conducted in March 2020). Title-Abstract-Keyword search was performed in Scopus while the Topic search was employed in all databases of Web of Science. The key search terms used in this review include "African Leaf Vegetable" OR "African leaf vegetables" OR "African vegetables" OR "Indigenous vegetables" OR "Traditional vegetables ".

### 2.3. Study Selection

All peer-reviewed articles or references from WoS and Scopus were exported to DistillerSR Evidence Partners Incorporated web-based applications. Duplicates were quarantined before conducting initial screening using the duplicate detection function. DistillerSR application allows for data extraction from included articles based on study characteristics.

### 2.4. Exclusion of Irrelevant Publications

The literature search results in both Scopus and WoS were screened based on different inclusion and exclusion criteria, including the language, type, geographical region, and focus of the publications (Table 1). Only the articles that were relevant after the full-text screening were considered for data extraction. The included articles' references were scanned for potential references that might not have been indexed in WoS and Scopus databases. The data from these articles were also added to the included articles for data extraction. These additional studies identified through hand picking were searched in Google Scholar.

**Table 1.** Inclusion and exclusion criteria.

| Variant | Inclusion Criteria | Exclusion Criteria |
|---|---|---|
| Language | Text in English | Text in languages other than English |
| Type | Publication type includes original research articles, review papers, books, or book chapters | Publication type is other than original research articles, review papers, books, or book chapters |
| Species | Relates to African Leafy Vegetables | Relates to crops other than ALVs |
| Region | Focuses on southern Africa | Focuses on areas other than southern Africa |
| Focus | Addresses at least one of the topics: utilisation, reliability of yield and marketing demand, production, production constraints, and breeding | Addresses none of the topics: utilisations, reliability of yield and marketing demand, production, production constraints, and breeding |

### 2.5. Data Extraction, Synthesising, and Reporting

The data extracted included the author's name, publication year, country, study participants, the species of ALV investigated, study design, analytical framework, and ALV attribute investigated. This review was summarised into specific themes that emerged from the included articles. The summarising stage also included supporting research evidence from grey literature. This was important to support the claim that more work needs to be done in this research area to understand consumer demand and the market value of ALVs. This was also important, especially in identifying future research directions and knowledge gaps, which form the rationale for conducting scoping reviews.

## 3. Results

### 3.1. Overview of Studies Identified

The search strategy identified 912 studies in the Web of Science and Scopus. These articles were then sent to the DistillerSR web-based application for screening. Duplications were removed, resulting in 676 unique articles. A total of 467 articles were removed using title screening, and a further 376 reports were excluded following the abstract screening. After examining the full texts of the remaining articles, 71 met the inclusion criteria (Figure 1).

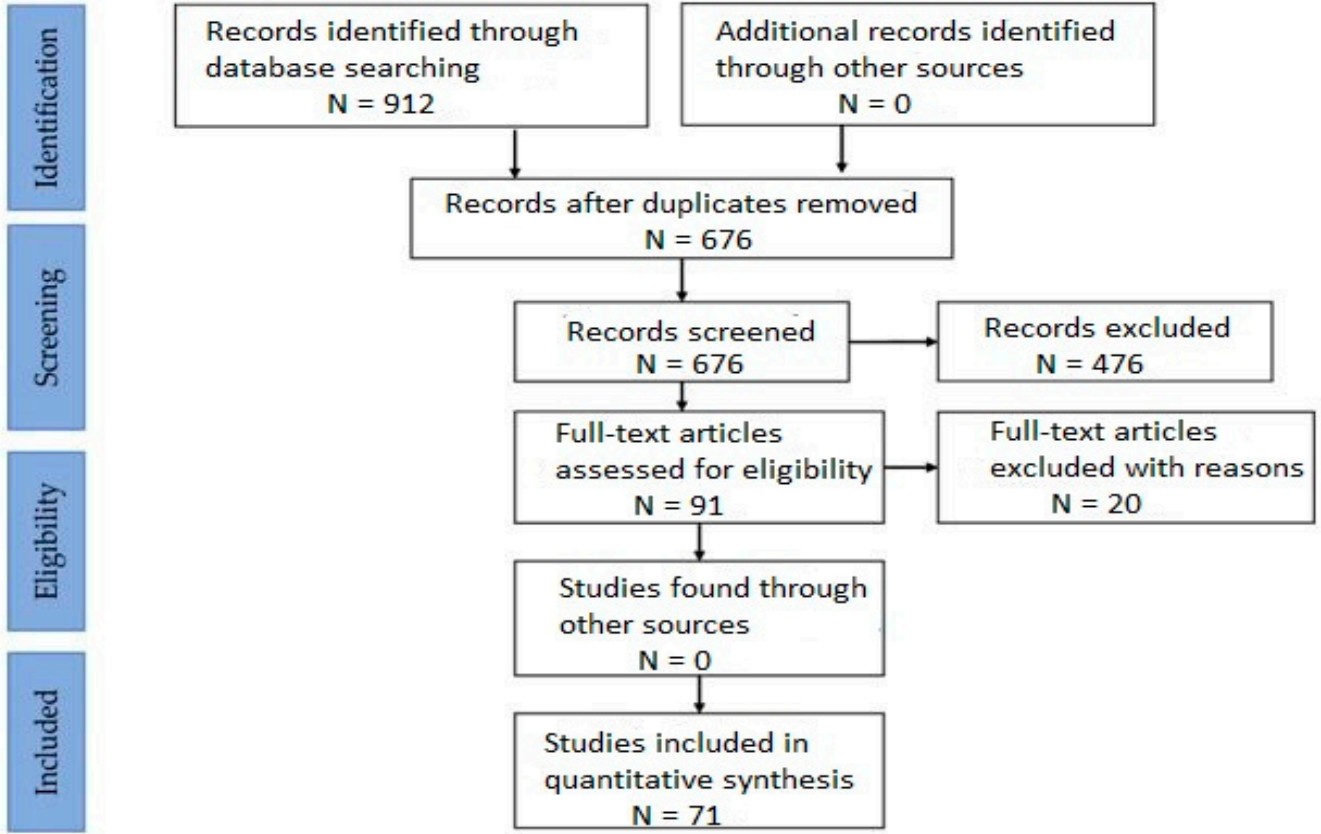

**Figure 1.** Flow chart of the selection of studies that investigated African leafy vegetables (ALVs) value chain components in southern Africa.

### 3.2. Publications on African Leafy Vegetables

The distribution of publications on ALVs across years, countries, and value chain components are shown in Figure 2. Figure 2a shows an increasing trend in the number of publications on ALV in southern Africa from 2000 to 2019. The highest increase in reports on ALVs was recorded in 2011 and 2017. Despite the emergence of the ALV research theme, few countries other than South Africa have been covered (Figure 2b). The entire 71 articles used in this study were derived from 5 countries (South Africa, Tanzania, Zimbabwe, Malawi, and Mozambique) out of the 12 nations constituting the southern African region. The bulk of the studies were for South Africa (76%), followed by Tanzania (15%), and Zimbabwe (4%). Malawi and Mozambique accounted for 1% of the entire panel of selected studies. The selected publications have addressed different aspects of the ALVs value chain, but most papers focused on producing, consumption, marketing, and processing the ALVs, respectively (Figure 2c).

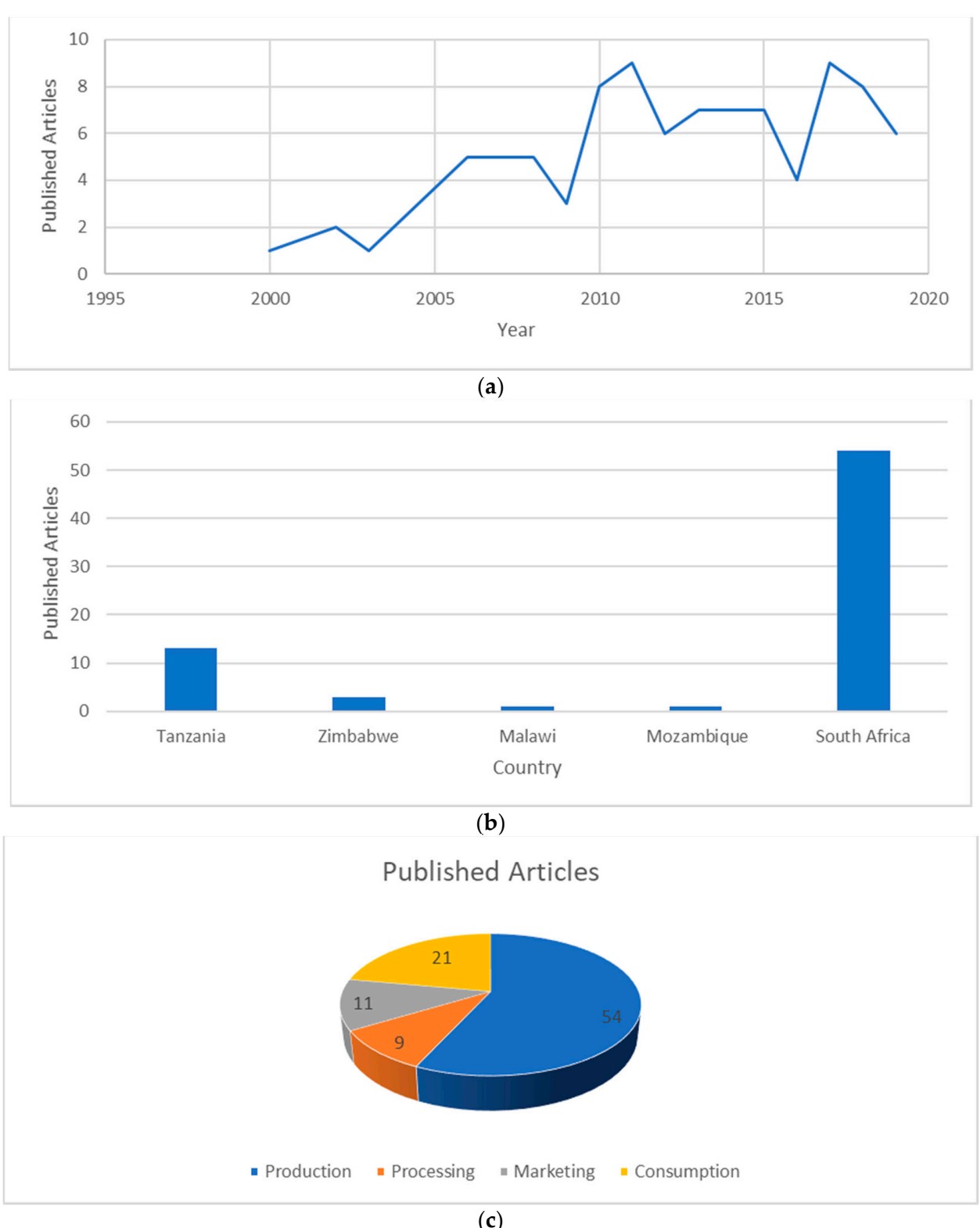

**Figure 2.** The trend in publications on African leafy vegetables: (**a**) evolution of the number of publications from 1995 to 2020; (**b**) distribution of publications across southern Africa; (**c**) value chain components investigated. All data are for the period shown in (**a**).

### 3.3. African Leafy Vegetable Value Chain Attributes

3.3.1. African Leafy Vegetable Production

The attributes of the 54 articles based on components of ALV production are summarised in Table 2. The study results show that significant interest in the production of ALV stems from their functional properties as 16 out of 54 of the production-oriented reports were based on nutritional aspects of the vegetables. The protein and micro-elements, including iron, calcium, magnesium, and copper, of some prominent ALVs such as *Amaranthus*, *Solanum nigrum*, and *Corchorus* spp. have been well characterised [3,11]. A single serving of an ALV based dish can provide 11.6–15.8 mg of iron and 1.4–3.7 mg of zinc. ALVs also contain antioxidants and phenolic compounds such as protocatechuic acid, hydroxybenzoic acid, etc. However, uncooked ALVs have anti-nutritional factors such as tannins, phytic acid, alkaloids, oxalic acid, and cyanogenic glycoside.

**Table 2.** Summary of ALV production components investigated.

| Production Component | Conclusion | Author |
|---|---|---|
| Breeding | Loss of biodiversity in *Amaranthus*, cucurbits and *Plectranthus esculentus* | [20] |
| | Genetic diversity in ALVs is yet to be explored | [21] |
| | A weak linkage exists among the value chain actors | [22] |
| | Rediscovery of an African vegetable *Plectranthus esculentus* | [23] |
| | ALV breeding strategy should involve pre-breeding, crossing, and participatory variety selections | [24] |
| | The paper identifies gene-banks for breeding lines and improved ALV cultivars | [25] |
| | There is wide phenotypic diversity conserved within Amaranthus species | [26] |
| | Considerable germplasm for pre-breeding exists for a wide range of ALV species | [27] |
| | The Agricultural Research Council of South Africa is a major ALV gene-bank in the region | [28] |
| | farmers prefer *Brassica juncea* and *Brassica rapa* varieties with delayed flowering and drought tolerance | [29] |
| | There is considerable genetic diversity within ALV species | [30] |
| | Yield and taste are the most preferred traits for *Amaranthus* and *Vigna unguiculata* | [31] |
| | Seeds of ALVs should be stored at 4 [±2] °C at 15–30% relative humidity | [32] |
| Breeding and Functional properties | *Amaranthus* accessions vary in protein content and agronomic traits | [33] |
| | Nutrition and access to certified seeds are drivers of ALV production | [34] |
| | *Amaranthus* species vary in mineral content suggesting the need for selection | [35] |
| Cropping systems | Upscaling of ALVs production will be met with several unexpected constraints | [36] |
| | To increase their productivity, farmers must expand the area under indigenous vegetable cultivation | [37] |
| | An in-depth investigation of the current status of ALV production practices is required | [38] |
| | Availability of relief packages negatively affect the production of ALVs in rural communities | [39] |
| | Most ALVs grow like weeds and are not cultivated | [40] |

**Table 2.** *Cont.*

| Production Component | Conclusion | Author |
|---|---|---|
| Cropping systems and Fertiliser regimes | Optimisation of Kraal manure application and leaf harvesting enhances the productivity of *Cleome gynandra* | [41] |
| Fertiliser regimes | The optimum rate of N application for high biomass accumulation in *Amaranthus*, *Vigna unguiculata* and *Corchorus* is 50 kg/ha | [42] |
| | Kraal manure and inorganic fertilisers combined result in high ALV biomass | [43] |
| | Sheep Kraal manure is ideal for *Amaranthus* in when inorganic fertilisers are unavailable | [44] |
| | Remote sensing can be used to predict fertiliser needs for ALVs | [45] |
| | *Amaranthus* may require 9.01 g/plant of N for optimum biomass accumulation | [46] |
| Functional properties | *Amaranthus*, *Galinsoga*, *Corchorus*, *Bidens Pilosa*, and *Cleome gynandra*, are rich in antioxidants and total phenolics. | [47] |
| | ALVs are valuable sources of phenolic compounds as compared to some exotic species. | [48] |
| | ALVs are rich in phenolic acids such as protocatechuic acid, hydroxybenzoic acid, etc | [49] |
| | Unprocessed ALVs contain anti-nutrients such as tannins, phytic acid, alkaloids, oxalic acid, and cyanogenic glycoside. | [50] |
| | Fried and boiled *Amaranth* leaves contained 627 and 429 μg retinol activity equivalents/100 g respectively. | [10] |
| | Eight AIVs had high iron, calcium and zinc contents | [51] |
| | *Solanum nigrum* and *Corchorus* are rich in microelements, including iron, calcium, magnesium and copper. | [52] |
| | mineral content in ALVs is much high than in most conventional exotic leafy vegetables | [11] |
| | Toxic metals such as cadmium can be exposed to humans via *Tulbaghia violacea* consumption | [53] |
| | The ALV dish contributed 11.6–15.8 mg Fe and 1.4–3.7 mg Zn | [54] |
| | ALVs are low in fat and high in folate which prevents chronic diseases | [13] |
| | Most of the ALVs are rich in vitamin A and, to a lesser extent, iron | [3] |
| | ALVs have pharmaceutical properties | [55] |
| Pest and disease control | PCR-based methods can detect and identify mycotoxins in ALVs based products | [56] |
| | Three subgroups exist among aphids species attacking *Amaranthus* and *Solanum nigrum* | [57] |
| | Arbuscular mycorrhiza fungus [AMF] affect the growth of *Cucumis myriocarpus* | [58] |
| | Mycotoxins are also present in ALVs | [59] |
| Planting methods | Seed priming with biostimulants reduces abiotic stress during the emergence of *Ceratotheca triloba* plants. | [60] |
| | Germination of *Brassica rapa*, *Citrullus lanatus* and *Solanum retroflexum* seeds is optimal when sown close to the soil surface. | [61] |
| | Seeds of indigenous vegetables should be sown at shallow depths to ensure a good plant stand | [62] |

**Table 2.** *Cont.*

| Production Component | Conclusion | Author |
|---|---|---|
| | Taro has potential for drought tolerance | [63] |
| | Stomatal conductance is critical in regulating moisture stress in the Taro | [64] |
| Water use efficiency | 30% ETc can be recommended for *Vigna unguiculata* and *Corchorus olitorius* suggesting drought tolerance | [65] |
| | AquaCrop models can be used for irrigation scheduling for sustainable ALV production | [66] |
| | ALVs have superior nutritional water productivity when compared to exotic vegetables | [67] |
| Water use efficiency and Functional properties | Application of 60% ETc enhanced nutritional content of ALVs while the concentration of nutrient under water stress indicated drought tolerance | [68] |

The selection of germplasm for conservation and identification of parental genotypes, usually referred to as pre-breeding, is the foremost step in plant breeding. In this study, 14 articles explored the genetic diversity within different ALV species. *Amaranthus* shows considerable variation for phenotypic traits, nutrition, and sensory qualities [26,31]. On the contrary, one study emphasises that genetic diversity in ALVs is yet to be explored. Another study reports a loss of biodiversity in *Amaranthus*, cucurbits, and *Plectranthus esculentus*. However, the World Vegetable Centre in Tanzania [25] and the Agriculture Research Centre in South Africa [28] are two major gene banks with extensive ALV accession collections useful for cultivar development. One study suggests that ALV cultivars' improvement should involve pre-breeding, crossing, and participatory variety selections [24].

Six studies explored cropping systems desirable for the cultivation of ALVs. Findings from two reports indicate that higher production is associated with an increase in area under cultivation of ALVs (rather than higher yields). In comparison, the availability of relief packages is associated with lower production. In addition, the upscaling of ALV production will be met by many obstacles, and chief among them is that most ALVs are 'volunteer' plants that grow like weeds [40]. For example, *Bidens pilosa* and *Amaranthus* spp. are common weeds in crop production systems.

The water use efficiency of ALVs was examined in six studies. The conclusions drawn from the studies highlight the considerable moisture stress tolerance of some selected ALVs. Two studies revealed the potential for drought tolerance in taro derived from its efficient stomatal conductance. Another study observed the highest water use efficiency when *Vigna unguiculata* and *Corchorus olitorius* were supplied with 30% etc., suggesting drought tolerance of these ALVs. Furthermore, AquaCrop models are useful for irrigation scheduling in ALV production [66].

Four studies explored pests and diseases affecting the production of ALVs. Two studies report that mycotoxins are present in raw and uncooked ALVs, including *Amaranthus*, and PCR tools can be used to detect mycotoxins in products [56]. Concerning pests, one study identified three subgroups of aphid species that attack *Amaranthus* and *Solanum nigrum.*

Plant nutrition and fertiliser recommendations required to achieve optimum growth in ALV production were summarised in six studies. N application's optimum rate for high biomass accumulation in Amaranthus, *Vigna unguiculata*, and *Corchorus* is 50 kg/ha [42]. Two studies recommend the integration of inorganic and organic fertiliser in ALV production systems. Where access to inorganic fertilisers is limited, sheep kraal manure is ideal for *Amaranthus.* About 9.0 g/plant of N is required for optimum biomass accumulation in *Amaranthus.* One study suggests using remote sensing tools to predict the fertiliser needs of all the other ALVs [45].

Two studies explored the planting methods for diverse ALVs. One of the studies recommends seed priming with biostimulants to reduce abiotic stress during the emergence of *Ceratotheca triloba* plants [60]. Another study suggests that the germination of *Brassica*

*rapa, Clematis lanatus,* and *Solanum retroflexum* seed are optimal when sown close to the soil surface [61].

### 3.3.2. African Leafy Vegetable Processing

The ALV processing methods reported in nine studies are summarised in Table 3. The analysis of processing factors showed that sun-drying, boiling, brining, blanching, and mechanised drying, respectively, were the main ALV processing methods investigated. Three studies reported the importance of sun-drying as a sustainable way of preserving ALVs among vulnerable groups. Sun-dried ALVs are a vital source of food and nutrition for many households during the dry season. Mechanised solar drying retains more nutrients during processing and is suggested as an effective way to provide access to nutritious vegetables, particularly in regions faced with frequent droughts. Boiling is also a common method used in the processing of ALVs for both immediate consumption and preservation. One study revealed that a boiling period of 5 min is adequate to remove the phytic acid content in 12 ALVs. However, the β-carotene content of fried *Amaranthus* was higher than that of boiled *Amaranthus* [10]. The ALV value chain actors need to build capacity in processing and preservation techniques to stimulate consumption and uptake of ALV produce [69]. Provision of processing and cold storage facilities to producers increases the value-added and profitability of ALVs [70].

**Table 3.** Summary of ALV processing components investigated.

| Processing Component | Target Group | Conclusion | Author |
|---|---|---|---|
| Sun-drying; Brining; Boiling | Rural consumers | There is a need for vigorous awareness campaigns to promote traditional vegetable use and nutritional education including proper cooking and preservation techniques | [69] |
| Sun-drying; Boiling | Rural consumers | Nutritional analysis of a broad spectrum of uncooked, cooked and preserved ALV dishes need to be done to determine their actual nutrient contribution. | [71] |
| Boiling | Unspecified | A boiling period of 5 min is adequate to remove the phytic acid content in 12 ALVs altogether. | [50] |
| Sun-drying; Brining; Boiling | Rural consumers; Urban consumers | The β-carotene content of fried Amaranth was higher than that of boiled Amaranth. | [10] |
| Sun drying | Rural consumers | Dried ALVs contribute significantly to the total vegetable consumption of the poorer households | [72] |
| Sun-drying; Mechanised drying | Rural consumers | Promotion of solar drying of ALVs is suggested as an effective way to continuously access nutritious vegetables, particularly in regions faced with frequent droughts | [73] |
| Blanching; Sun-drying; Boiling | Peri-urban farmers; Commercial farmers; Rural consumers; Urban consumers; Traders | Promotion of processing ALVs for value addition including cold storage facilities nearer to the smallholder farmers will enhance profitability | [70] |
| Blanching; Sun-drying; Boiling | Rural farmers | Boiling (cooking) is the predominant processing method among ALV farmers | [74] |
| Sun-drying; Boiling | Rural consumer | Drying and storage methods for use during off-season periods should be optimised to minimise nutrient losses. | [75] |

### 3.3.3. Marketing of African Leafy Vegetables

The main factors affecting the marketing of ALVs studied among the included articles were market potential, willingness to pay, product distribution, and consumer attitudes, respectively (Table 4). Findings from most of the articles showed that the market potential of ALVs was of prime importance and high rural-urban migration presents an opportunity for the establishment of formal ALV markets. Some reports suggest a lucrative market for ALVs in informal markets [10]. It was also highlighted that product positioning strategies would enhance the visibility of ALVs in the formal markets leading to an established market share. At the same time, smallholder farmers can increase their gross margins through linkage with all the value chain actors [70].

**Table 4.** Summary of ALV marketing components investigated.

| Marketing Component | Target Group | Conclusion | Author |
|---|---|---|---|
| Potential market | Rural farmer; commercial farmer; traders | The issue of market differentiation is a significant bottleneck to the success and scaling-up of ALV seed systems | [76] |
| Potential market | Rural farmer; commercial farmer | Niche marketing of high-value ALVs will stimulate production | [78] |
| Potential market | Rural and Urban consumers; trader | There is a need for value addition of ALVs to position them for high-value niche market segments | [22] |
| Potential market | Rural farmer; rural and urban consumer; traders | The trading of ALVs in informal urban markets suggest the potential for commercialisation | [10] |
| Potential market; Product distribution | Rural farmer; rural and urban consumer; traders | Gender affects market performance and profitability of the ALV product value chain | [77] |
| Potential market; Willingness to pay | Rural farmer | Infrastructure support will enable ALV farmers to improve participation in markets and increase profits. | [79] |
| Potential market | peri-urban farmers; | Critical issues for ALV farmers relate to insufficient designated market space, insufficient information regarding the volumes and prices of particular produce in the market at any time. | [38] |
| Potential market; Willingness to pay | Rural farmer; rural and urban consumer; traders | Smallholder farmers currently make the highest gross margins, and more returns can be realised through linkage of all the value chain actors | [73] |
| Potential market; Product distribution; | backyard gardeners; Urban consumers; Traders | The market for ALV is insufficient, but demand can be boosted through product positioning strategies | [80] |
| Potential market | Rural farmers | High rural-urban migration presents an opportunity for the establishment of a formal market for *Brassica rapa* and *Solanum* spp | [74] |
| Potential market; Consumer attitudes; Willingness to pay | Urban consumers; Traders | Willingness to purchase ALVs diminishes with an increase in household income among Africans | [32] |

However, some studies highlighted the barriers to the market penetration of ALV products in markets [76]. The major problem was that small-scale farmers who produce ALVs lack access to market and market information on demand and supply for ALVs [38]. An additional setback to the success and scaling-up of ALV products is their general lack of product differentiation, i.e., little effort is made to distinguish ALVs in the market to make them more appealing to consumers. Two studies suggested that positioning as a high-value niche (e.g., nutritional benefits) would stimulate demand for ALVs.

Willingness to pay and consumer attitudes towards ALV products were also explored. One study reported that the desire to purchase ALVs diminishes with increased household income [32]. Another observed that gender affects the market performance and profitability of the ALV product value chain [77]: Women make critical decisions about the production

of ALVs for household consumption and trade in local markets while men are more active in the formal marketing of ALVs, particularly in high premium niche markets such as hotels and restaurants.

### 3.3.4. Consumption of African Leafy Vegetables

The analysis of factors affecting the consumption and utility of ALVs was investigated in 22 studies (Table 5). Of these, most studies assessed preference for particular ALV species. The most popular ALV species in Malawi include *Amaranthus*, *Brassica carinata*, mustard, and *Biden pilosa* [22], whereas *Solanum macrocarpon*, *Amaranthus*, *Corchorus olitorius*, *Brassica carinata*, and *Cucurbitaceae* spp. are important ALVs in Mozambique [22]. A study conducted in Zimbabwe revealed that consumers knew of about 79 ALVs that were fit for human consumption [49]. In Tanzania, consumers prefer ALVs such as *Vigna unguiculata*, *Manihot esculenta* (leaves), *Corchorus olitorius*, *Bidens pilosa*, and *Ipomoea batatas* (leaves) because of their long shelf life. Studies conducted in South Africa identified 33 commonly consumed ALVs [69], and *Amaranthus* spp, *Cleome gynandra*, *Citrullus lanatus*, and blackjack are the most preferred ALVs [32]. However, one of the reports noted that there was a general reduction in the availability of ALVs, which led to reduced consumption of both fresh and dried vegetables [71]. Ten studies explored the influence of demographic factors as a determinant of the consumption of ALVs. Three of the reports reveal the role of women as key players in the ALV value chain as they are more intensively engaged in the production and informal marketing and responsible for the bulk of the ALVs consumed in households. Hence, women should be given more support to improve the ALV value chain [78]. Sensory attributes of different ALVs are critical to the acceptability and consumption of certain ALV species among children [54]. The reports that explored the consumption patterns of ALVs also showed that rural societies were the primary consumers of ALVs. Six studies revealed the differences in the main use of ALV across regions. In Tanzania, ALVs are valued for their nutritional and medicinal properties [81]. In South Africa, ALVs serve many purposes; for example, one study reported that *Amaranthus* was mainly used as a leaf vegetable, livestock feed, and snuff [82]. Additional reports suggest the need to evaluate the medicinal properties of some ALVs.

**Table 5.** Summary of ALV consumption components investigated.

| Consumption Component | Country | Target Group | Conclusion | Author |
|---|---|---|---|---|
| Species preference | Malawi; Mozambique | Rural and Urban consumers; trader | Amaranthus, *Brassica nigra*, and *Bidens pilosa* were the most important ALVs in Malawi, while *Solanum macrocarpon*, *Amaranthus*, *Corchorus*, *Brassica nigra*, and *Cucumis anguria* are important ALVs in Mozambique | [22] |
| | Zimbabwe | Rural consumers | People of the Buhera district in Zimbabwe had knowledge of 79 ALVs | [49] |
| | South Africa | Rural consumers; Urban consumers | *Amaranthus* accession differ significantly in sensory qualities including taste | [83] |
| | South Africa | Unspecified | Consumption of ALVs such as *Amaranthus*, *Corchorus*, *Brassica rapa*, *Solanum nigrum*, *Cleome gynandra*, *Cucurbit maxima*, *Citrullus ecirrhosus*, and *Vigna unguiculata* should be promoted in South Africa | [84] |
| | Tanzania | Rural farmers | ALVs such *Vigna unguiculata*, *Manihot esculanta*, *Corchorus*, *Biden pilosa*, and *Ipomea batatas* are valued for their long shelf life | [85] |
| | South Africa | Rural farmers; Rural consumers | Thirty-three (33) traditional vegetables were identified among the respondents | [69] |
| | South Africa | Rural farmers; Rural consumers | There is reduced availability of ALV, which leads to reduced consumption of both fresh and dried African vegetables. | [71] |

<div align="center">**Table 5.** *Cont.*</div>

| Consumption Component | Country | Target Group | Conclusion | Author |
|---|---|---|---|---|
| Species preference; Demographic influence | South Africa | Urban consumers; Traders | Popular ALVs include *Amaranthus*, *Cloeme gynandra*, *Solanum* spp. The preference and utilisation of these ALVs varied among agro-ecological zones and ethnic groups. | [32] |
| Species preference; Demographic influence; Main use | Tanzania | Rural farmers | There were gender disparities in the frequency of consumption of ALVs, and the preference of some ALVs to others was determined by medicinal properties and ease of processing | [81] |
| Demographic influence; Main use | Tanzania | Rural consumers | Most households purchase and consume the open sun-dried types due to a lack of knowledge of solar-dried vegetables' benefits. | [86] |
| Demographic influence | South Africa | Rural farmers; Rural consumers; Urban consumers | Adult females are responsible for the high percentage of ALVs consumed in households | [21] |
| Demographic influence | Tanzania | Rural farmers | Female-headed households are more intensively engaged in ALV production. Hence female farmers need support to upgrade the value chain | [77] |
| | South Africa | Rural consumers | There is a general decline in the use of ALVs, and consumption of edible parts varies from one region to the other | [70] |
| | South Africa | Rural farmers; Rural consumers; Urban consumers | Female producers prefer supplying the demand of final consumers, while male traders are more involved in wholesale to restaurants and grocery stores. | [78] |
| | South Africa | Rural farmers; Rural consumers; Urban consumers; Traders | ALVs are consumed mainly by older people based in rural areas, those in urban areas who are aware of the taste and nutrition attributes of ALVs. | [73] |
| | South Africa | Rural consumers | Sensory attributes are essential determinants of the acceptability of AVLs among young children | [54] |
| | Tanzania | Rural farmers | There are marked regional and gender-based differences in ALV preferences that need to be noted for targeting future interventions | [34] |
| | South Africa | Rural consumers | Knowledge of edible ALV plants among parents and acceptance of the ALV based dishes among children indicate a potential for widescale adoption | [75] |
| Main use | South Africa | Rural consumers | Amaranth is mainly used as a leaf vegetable, while a few use it as livestock feed and snuff. | [82] |
| | South Africa | Unspecified | Consumption of ALVs can be enhanced by the combined efforts of agriculturalists and nutritionists | [87] |
| | South Africa | Urban consumers | ALVs should be considered as a source of both food and medicine | [55] |

## 4. Discussion

This review is the first study to employ the scoping methodology in understanding the state of knowledge and research evidence on the utilization of underutilised crop species. The study summarised and synthesised the results from 71 studies investigating the ALVs value chain's current status among countries in the southern African region. ALVs are an essential component of many southern Africans' diets, particularly among the poorer households. Despite the significance of ALVs in providing food and nutrition security, the entire 71 articles used in this study were derived from only 5 countries out of the 12 countries that make up the mainland southern Africa region (Figure 2b). This is surprising given the importance of the ALVs to the poverty-stricken general populace's food and nutrition. South Africa has been at the forefront of supporting research on

neglected and underutilised crop species. The Water Research Commission of South Africa, the Department of Science and Technology, and the Agriculture Research Council are among the research bodies providing funding support for the upscaling of ALVs in South Africa [18]. Tanzania also recorded many reports in this study, mainly because of the World Vegetable Centre, which has its regional office in the country [25]. One of the mandates of the World Vegetable Centre is the improvement and promotion of traditional African vegetables. Despite potentially having the highest consumption per capita of ALVs, other southern African states lack policy towards the research and funding to support underutilised crop species.

The results show that the production component of ALVs was the most studied (Table 2). Evidence from the 54 articles on production shows that the nutritional value of ALVs was an important factor in the decision to produce. This was expected because ALVs are valued for their nutrition and have been fundamental to many communities' nutritional security. Some ALVs such as *Amaranthus, Corchorus* spp., and *Solanum nigram* are rich in essential macro and micronutrients, including minerals, vitamins, quality vegetable protein, and dietary fibre [51]. Additionally, ALVs are rich in antioxidants, enhancing blood levels and combatting cancers, among other chronic diseases [47]. However, despite extensive literature describing the nutritional value of ALVs, there is a dearth of information regarding these nutrients' bioavailability. Bioavailability refers to the proportion of the nutrients that are absorbed and utilised by the body. The stable-isotope mass spectrometry can be recommended among other rigorous techniques useful for bioavailability assays in ALVs.

Cultivar development depends on three processes: Selection, breeding, and testing. Selection identifies superior genotypes for hybridisation; breeding recombines superior parental genotypes to create novel genetic variability; testing exposes the new candidate genotypes to the representative environments. Our results show that most studies were only concerned with the selection phase of the germplasm development pipeline. There were hardly any attempts to broaden the genetic base of any ALV through conventional breeding approaches. This explains the undeveloped ALV seed supply sector in the region. The non-availability of improved seed remains a significant impediment to the upscaling of ALVs [22], but there have been few attempts by local breeding programs to develop superior ALV germplasms. Access to good-quality ALV planting materials is a significant problem for farmers. Currently, farmers rely on landraces and volunteer plants as propagules. The use of elite germplasm is a prerequisite for high-quality produce; there is a need to develop good quality seed through breeding to make good quality products available for processing and marketing. Viable informal and formal indigenous vegetable seed supply systems exist in Kenya [9]. Considering the self-pollination mode and inconspicuous flowers of most ALV species, broadening the species' genetic base could be of great importance for the efficient and successful development of improved cultivars. Genetic variability can be enhanced through intra-specific crosses, mutation breeding techniques, and the introduction of germplasm from diverse geographical regions.

Understanding the best practices for the production of ALVs will increase the productivity and quality of indigenous vegetables. The response of ALV species to fertiliser and water requirements was explored by a few of the selected studies with some conclusive results. Yields and nutrient content of exotic vegetables, including *Brassica* spp., improves with an increase in the supply of macro and micronutrients, and the same can be envisaged about ALVs. Some ALVs such as the *Vigna unguiculata* have exceptional drought and heat tolerance [88]. It is also noteworthy that most ALV species can grow under very high plant densities. This competitive ability often leads to their classification as weeds, with some species such as the *Bidens pilosa* being candidates for the noxious weed group. However, the ability to thrive under high population densities may suggest drought tolerance. Pests and diseases also have a significant effect on vegetable production and productivity. However, the study results showed that very little is known about the extent of the susceptibility of most ALVs to pests and disease, although it is widely accepted that most of the ALVs

harbour pests and disease. *Solanum* spp. and most cucurbits are known to be vectors of blights and powdery mildews, which are detrimental to most susceptible exotic vegetables. Therefore, agronomic research initiatives should address plant densities, water use efficiency, allelopathic interactions, and yield variation when combinations of ALVs are to be integrated into existing cropping systems.

The study also revealed important findings regarding the status of the processing of ALVs in southern Africa (Table 3). There is a need to devise appropriate ALV processing techniques to maximise nutrient retention and prolong shelf life. The development of such processing systems requires knowledge of the current practices and how they influence product quality. Although most reports on processing included in the present study were cross-sectional, sun-drying and boiling were the most explored processing methods. This suggests that traditional processing methods were still essential in the preservation of ALVs. This indicates that ALVs are processed and used locally by families for home consumption and local sale. Farmers can increase income from ALVs by using appropriate processing technologies that add value to the finished products. Research into the development of appropriate processing technology must have an emphasis on low-cost technologies that are suitable at the village level to allow rural people the opportunity to generate income. Undoubtedly, the technologies applied to exotic vegetables can also be adapted to indigenous vegetables. Canning and freezing are among some of the techniques with the potential to add value and position ALVs for retail markets.

However, the study shows that the market analysis of ALVs in southern Africa is poorly documented, and the only evidence available is based on 10 cross-sectional reports (Table 4). The market potential, willingness to pay, product distribution, and consumer attitudes were the explored factors in the analysis of the marketing of ALVs. Generally, ALVs are traded in informal markets across the southern African region, and minimal success has been achieved in the commercialisation of indigenous vegetables [22,39]. Extensive market research and development are essential if substantial commercial interests in ALVs are to emerge beyond the current roadside and local markets. Market research will identify strengths, weaknesses, and opportunities in the vegetable market segments and gather information about the business environment. Through this exercise, impediments to the marketing of ALV based products can be identified. The goal is to gather information from all stakeholders involved in the marketing of the products. According to the opportunities identified, studies of the other areas of enterprise development are then undertaken.

The lack of access to demand and supply information was identified as the major constraint affecting ALVs enterprise growth in the region [38,89]. For local farmers to be competitive in informal and formal markets, they need to monitor market demand and market prices. If ALV product models are determined by market dynamics alone, then the vegetables' economic importance will remain compromised. Hence there is a need to develop market linkages among farmers and all the stakeholders in the value chain for strategic alliances inproducer–industry partnerships. Public and private sector support can facilitate linkages between farmers and retailers by establishing farm consortiums, ground rules for farmer-buyer contracts, and sharing experiences from successful marketing schemes. For example, in east Africa, the World Vegetable Centre and Farm Concern International collaborated to upscale the utilisation and marketing of ALVs and streamline the ALV value chain's efficiency for income generation among the vulnerable groups in Kenya and Tanzania [90]. The initiative established contractual agreements between farmers and chain stores such as Uchumi, Nakumatt, and Tuskys in Kenya. Whereas in Tanzania, intervention came through capacity building in value addition based on processing technologies for the offseason markets. In the end, annual profits per 0.1 ha of ALVs such as *Amaranthus*, *Solanum nigrum*, and *Vigna unguiculate* were valued at US \$4571, \$3033, and \$2426, respectively, compared to \$1760, \$1400, and \$960 for kale, spinach, and cabbage, respectively [90]. Farmers in southern Africa can benefit from similar linkages given the well-established networks of chain stores throughout the region.

Besides nutritional benefits, consumers value ALVs for their nutritional and medicinal benefits (Table 5), suggesting further market potential and the need for product positioning strategies. The high nutritional value and therapeutic properties of some ALVs can be exploited by producing products for niche markets, such as health foods or natural products. A good example is the significant impact of Asian imported *Aloe vera*-based products on the health-conscious niche throughout the African continent.

## 5. Limitations

The study confirms the scoping review methodology's utility for priority setting in the upscaling of indigenous underutilised crop species. Many knowledge gaps have been identified despite the vast terminology used to classify ALVs. Several relevant articles, reports, and grey literature, may have been missed in this study. Many ALVs are multi-purpose crops that can also fit as fruits or pulses. Hence studies that did not emphasise their importance as vegetables may even have easily been obscured. The shortage of quantitative studies meant that most encountered studies on ALVs were qualitative, including surveys and case studies. Nonetheless, scoping reviews are not meant to assess the quality of literature scoped. Future studies on well-covered aspects such as the different ALV species' functional properties and consumer attitudes, such as willingness to pay, may consider empirical methodologies.

## 6. Conclusions

A scoping review was conducted to evaluate the ALVs value chain's current status in the southern African region. The ALVs have high nutritional value and are essential in the food and nutrition security of rural, peri-urban, and urban communities. However, the commercial success of ALVs remains challenged by many constraints. This study identified a set of innovative research and development efforts across the entire value chain that could scale-up and mainstream ALVs in the region. A viable seed supply system will require extensive ethnobotanical surveys and germplasm collection, participatory varietal selections, germplasm characterisation using morphological and molecular markers, parental selection and crosses, integration of marker-assisted breeding, and stability tests. Investments in low-cost value addition machinery for solar drying, canning, and freezing will increase the competitive advantage of ALV based products. The enhanced performance in the marketing of ALV based products would transform the lives in rural economies. Efforts should be channeled at informing consumers about the benefits of ALVs (to create demand), supporting farmers by linking them with markets to ensure supply, and providing supportive policies to facilitate the strategic positioning of ALVs.

**Author Contributions:** Conceptualization, T.M. and A.I.T.S.; methodology, A.I.T.S. and T.M.; formal analysis, A.I.T.S.; investigation, all authors; resources, T.M. and G.M.M.; data curation, A.I.T.S.; writing—original draft preparation, A.I.T.S. and T.M.; writing—review and editing, all authors; visualisation, A.I.T.S.; supervision, T.M. and A.T.M.; project administration, T.M. and G.M.M.; funding acquisition, G.M.M., S.M., M.M., and T.M. All authors have read and agreed to the published version of the manuscript.

**Funding:** This research was funded by the Global Challenges Research Fund (GCRF) Interdisciplinary Research Award 2019, [Grant number RIS 2421450/RIS2094206]. The Water Research Commission of South Africa is acknowledged for funding the through WRC Project Number K5/2717//4 on 'Developing guidelines for estimating green water use of indigenous crops in South Africa and estimating water use using available crop models for selected bio-climatic regions of South Africa'.

**Institutional Review Board Statement:** Not applicable.

**Informed Consent Statement:** Not applicable.

**Data Availability Statement:** The data presented in this study are available on request from the corresponding author.

**Conflicts of Interest:** The authors declare no conflict of interest.

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
