# Peer review of "African Leafy Vegetables for Improved Human Nutrition and Food System Resilience in Southern Africa: A Scoping Review"

_sustainability, doi:10.3390/su13052896_

Round 1

Reviewer 1 Report

Reviewer 1

Review of African Leafy Vegetables for improved human nutrition and food system resilience in southern Africa: a scoping review, Shayanowako et al

Brief summary - The paper presents the findings of a scoping study that drew on published literature in relation to the production, processing, marketing and consumption of African leafy vegetables (ALVs). The authors reviewed the published material to identify gaps in knowledge related to ALVs and proposed a number of areas of research that would assist in greater production and consumption of these crops for the benefit of growers and the consuming public.

General comments - The article is well researched and presented and generally easy to read. The Discussion tends to be a little repetitious of material covered elsewhere in the article and could probably be reduced in length without detracting from the findings. Further, although the authors discuss various knowledge gaps and opportunities for future research in the Discussion, it would be useful to collate those opportunities in an easy-to-read table, and perhaps even offer some suggestions for prioritisation.

Specific comments

Line 40 – ‘custom, habit, or tradition’ – are not these all the same thing?

Line 56 – ‘even before the harvest of the earliest legume’ – what does this mean? I suspect it is a reference to some stage of the growing season, wherein the ALVs are ready for harvest before early legume crops, but there could perhaps be some re-wording or additional information to clarify for those not familiar with the growing environment and production systems of southern Africa.

Line 57 – ‘vulnerable groups during the 'hungry period'.’ Some definition for clarity would be useful. I suspect ‘vulnerable groups’ are small holder farmers and the rural poor, and the ‘hungry period’ is the late dry season, but once again, some additional descriptors would help clarify for readers who are not familiar with the socio-economic or agri-climatic characteristics of the region of study. If there is reluctance to identify the groups that represent those who are ‘vulnerable’, perhaps a generic description, such as ‘those suffering food security’ or similar would suffice.

Line 97 – in keeping with other citation styles, I suspect the author of 18 should not be listed here.

Line 139 – it may be useful to provide a list of the SADC states, either in the text or as a table. Fig 2b gives an insight to some of them, but it is elsewhere noted there are 16, but no indication of what they are.

Line 181 – not convinced that the last box in Fig 1 is required, since it only repeats the previous box.

Line 200-208 – I question whether nutrition-related information is appropriately covered under the topic of ‘production’. It almost deserves a section in its own right, as it relates to the dietary functionality of the plants, rather than production. Other production related aspects such as germplasm, farming systems, fertilisation etc. are appropriately covered in this section.

Line 239 – ‘Another study recommended 30% ETc for Vigna 239 unguiculata and Corchorus olitorius, suggesting the drought tolerance of these ALVs.’ – not sure what this means. Does it mean that these crops can survive on a water supply that is 30% of their ET demand, or that they should be under-irrigated by 70%? Some re-wording to provide clarity would be useful.

Line 244 – ‘and PCR tools can be used to detect mycotoxins in products’ – this may well be true, but is it really relevant to ALVs? The technology can be used to detect mycotoxins in many products, presumably.

Line 270 – needs reference citation added.

Line 271 – some repetition, needs editing

Line 311 – section 3.3.4 is a repeat of the first paragraph of section 3.3.1 and is not about consumption of ALVs. I suspect that section 3.3.4, when re-written to cover the topic of consumption, should refer to Table 5, which is currently mentioned only in the Discussion.

Line 402 – ‘allow rural people the opportunity to generate.’ Generate what?

Reviewer 2 Report

The research article entitled “African Leafy Vegetables for improved human nutrition and food system resilience in southern Africa: a scoping review” addresses a systematic review assembled and examined scattered knowledge generated on ALVs across southern Africa, with a particular focus on the production, processing, marketing, and consumption.

The scope of the study is of extreme importance, as knowledge of the current status of ALVs in the region is of major importance to identify current challenges and constraints to address further studies regarding food and nutrition security. Overall, the ms is well-written, with minor English needed revision, and the objectives are stated. However, I have some suggestions about the Introduction, methodology, and discussion. Through the text, it is somewhat confusing if the analysis is done solely at the Southern African States and then narrowed to South Africa. Please be more thorough throughout the text. Please find major comments below and minor issues in the attached file.

In the Introduction, I would suggest first start the text from African leafy vegetables and then narrow to Southern Africa.

L41- 42: Do you have any production trend data on Amaranthus spp., Solanum spp., Cleome gynandra, Vigna unguiculata, Bidens pilosa, and Corchorus olitorius are among the most popular cultivated ALVs? Have you searched at FAOSTAT data?

L57- “hungry period”- please develop this idea, when is this period and why.

L71- “Despite the potential of ALVs to contribute to food and nutrition security”- please explain why the ALVs contribute to nutrition security, as for food security it is well explained in the paragraph above.

L79-80 “The lack of improved cultivars and continued use of landraces means that productivity 79 remains low (yield per unit land). “- please add reference and example in comparison to standard crop yields.

M&M

2.1. Please include the ultimate research question addressed.

L139- “Southern Africa Development Committee (SADC) states.”- please add a reference to know which states are included.- maybe adding a figure would help the readers to easily follow the rest of the ms.

Results

L178, please include the excluding criteria in the results description for titles and abstracts.

At the discussion level, I believe that authors could include a section regarding the importance of studies for nutrition and food security, to address fully the title of the ms, and its ultimate objective.

Round 2

Reviewer 2 Report

The authors have significantly improved the manuscript and have addressed positively to all comments.

The manuscript is an excellent contribution to the field of research and thus i recommend its publication at Sustainability in its present form.

Author Response

we acknowledge the comments. Thank you